# Extreme weather events and dengue in Southeast Asia: A regionally-representative analysis of 291 locations from 1998 to 2021

Sophearen Ith[1,2], Xerxes Seposo[3,4]*, Vitou Phy[5], Kraichat Tantrakarnapa[6], Geminn Louis C. Apostol[4], Pandji Wibawa Dhewantara[7], Rozita Hod[8], Mohd Rohaizat Hassan[8,9], Hidayatulfathi Othman[10], Mazrura Sahani[10], Jue Tao Lim[11], Ha Hong Nhung[12], Nguyen Hai Tuan[12], Ngu Duy Nghia[12], Taichiro Takemura[13], Inthavong Nouhak[14], Paul L. C. Chua[1], Alex R. Cook[15], Felipe J. Colón-González[16,17,18], Masahiro Hashizume[1]

1 Department of Global Health Policy, Graduate School of Medicine, The University of Tokyo, Tokyo, Japan, 2 Department of Epidemiology, National Institute of Infectious Diseases, Japan Institute for Health Security, Tokyo, Japan, 3 Department of Hygiene, Graduate School of Medicine, Hokkaido University, Hokkaido, Japan, 4 Ateneo Center for Research and Innovation, Ateneo de Manila University-School of Medicine and Public Health, Pasig, Philippines, 5 Indeed Inc, Tokyo, Japan, 6 Department of Social and Environmental Medicine, Faculty of Tropical Medicine, Mahidol University, Bangkok, Thailand, 7 Research Center for Public Health and Nutrition, National Research and Innovation Agency, Bogor, Indonesia, 8 Department of Public Health Medicine, Faculty of Medicine, Universiti Kebangsaan Malaysia, Cheras, Kuala Lumpur, Malaysia, 9 University of Cyberjaya, Cyberjaya, Selangor, Malaysia, 10 Centre for Toxicology and Health Risk Studies, Faculty of Health Sciences, Universiti Kebangsaan Malaysia, Kuala Lumpur, Malaysia, 11 Lee Kong Chian School of Medicine, Nanyang Technological University, Singapore, Singapore, 12 National Institute of Hygiene and Epidemiology, Hanoi, Vietnam, 13 Institute of Tropical Medicine, Nagasaki University, Nagasaki, Japan, 14 Lao Tropical and Public Health Institute, Ministry of Health, Vientiane, Laos, 15 Saw Swee Hock School of Public Health, National University of Singapore and National University Health System, Singapore, Singapore, 16 Data for Science and Health, Wellcome Trust, London, United Kingdom, 17 Department of Infectious Disease Epidemiology, Faculty of Epidemiology and Population Health, London School of Hygiene & Tropical Medicine, London, United Kingdom, 18 School of Environmental Sciences, University of East Anglia, Norwich, United Kingdom

* seposo.xerxestesoro@pop.med.hokudai.ac.jp

## Abstract

### Background

Climate change, leading to more frequent and intense extreme weather events (EWEs), could significantly impact dengue transmission. However, the associations between EWEs and dengue remains underexplored in the Southeast Asia (SEA) region. We investigated the association between selected EWEs (i.e., heatwaves, extremely wet, and drought conditions) and dengue in the SEA region.

### Methods and findings

Monthly dengue case reports were obtained from 291 locations across eight SEA countries between 1998 and 2021. Heatwaves are defined as the monthly total number of days where temperatures exceed the 95th percentile for at least two consecutive days. Droughts and extremely wet conditions are defined by a self-calibrating Palmer Drought Severity Index (scPDSI). We implemented a generalized additive

**Data availability statement:** Dengue data of Cambodia can be freely downloaded from Project Tycho (https://doi.org/10.25337/T7/ptycho.v2.0/KH.38362002). Climate data can be downloaded from Copernicus (https://doi.org/10.24381/cds.68d2bb30). The self-calibrating PDSI can be sourced from the Climate Research Unit (https://crudata.uea.ac.uk/cru/data/hrg/). Population data can be downloaded from the Socioeconomic Data and Application Center (https://doi.org/10.7927/H4PN93PB). We are unable to share the dengue case data used in this study because, except for the data from Cambodia, all other dengue case data were obtained through a collaborative platform under the Southeast Asia Research on Climate Change and Dengue (SEARCD) initiative. This data was accessed under a formal data-sharing agreement, which restricts public dissemination. The dataset was acquired with explicit permission from SEARCD members representing each participating country, and any further sharing is subject to the agreement's terms and conditions. The data sets and the analysis scripts may be accessed upon formal request by contacting the Department of Hygiene, Graduate School of Medicine, Hokkaido University, via email at eisei@med.hokudai.ac.jp.

**Funding:** This work was funded by the Japan Society for the Promotion of Science Early-Career Scientist Grant (22K17374) awarded to XS, Start-up Grant awarded to SI (24K23711) and the Japan Science and Technology Agency (JST) as part of SICORP (JPMJSC20E4) awarded to MH. The funders had no role in study design, data collection and analysis, decision to publish, or preparation of the manuscript.

**Competing interests:** The authors have declared that no competing interests exist.

mixed model coupled with a distributed lag non-linear model to estimate the association between each EWE and dengue. Months with fewer than 12 heatwave days increased dengue risk with delayed effect after two months lag, compared with months without any heatwave. Highest dengue risk is at 7 heatwave days (RR = 1·28; 95%CI: 1·19,1·38). Compared to normal conditions (i.e. scPDSI = 0), drought conditions (i.e. scPDSI = −4) were positively associated with dengue risk (RR = 1·85; 95%CI: 1·73,1·99), while extremely wet conditions (i.e. scPDSI = 4) have reduced dengue risk (RR = 0·89; 95%CI: 0·87,0·91). Although the findings of this study are significant, its limitations arise from the inconsistency of dengue case reporting, which might complicate dengue risk estimation.

## Conclusions

This study shows that the delayed effect of heatwaves and drought conditions magnifies the risk of dengue in the SEA region. The findings highlight the need for public health interventions to mitigate the potential dengue risks posed by EWEs in the context of climate change in SEA. Future research should investigate the factors influencing variations in the EWE-dengue association across the region to support the development of tailored, location-specific mitigation and prevention strategies.

## Author summary

Dengue remains a major public health concern in Southeast Asia region, where the warm and humid climates create optimal conditions for dengue transmission. Extreme weather events are expected to intensify with climate change and may further influence dengue disease. In this study, we examine the association between extreme weather events—such as heatwaves, droughts, and extremely wet conditions—and dengue, using data from 291 locations across eight Southeast Asian countries from 1998 to 2021. We found that heatwaves were associated with an increased risk of dengue after a two-month lag. Drought conditions showed a significant positive association with dengue. Interestingly, while extremely wet conditions are linked to lower dengue risk, very high wetness shows a non-significant increase, requiring further investigation to better understand these patterns. These findings highlight the critical role of extreme weather events in shaping dengue transmission and provide valuable insights for targeted public health interventions. By understanding these weather-dengue relationships, policymakers can strengthen preparedness and mitigate the impacts of climate-sensitive diseases like dengue.

## Introduction

Dengue is caused for flaviviruses, transmitted by the infectious bite of *Aedes aegyti,* and *Ae. albopictus* mosquitoes. Dengue poses an alarming impact on human health,

and the global economy, particularly in the Southeast Asian (SEA) region [1]. The climate in the SEA region, characterized by persistent hot and humid weather throughout most of the year, offers optimal environmental conditions for mosquito development and dengue transmission [2]. According to a recent projection study, dengue incidence in SEA is expected to reach its peak in the 21st century [3]. By the end of this century, the length of dengue transmission season in the SEA region is expected to be longer with a rise in number of populations at risk of up to 696 million additional people [4].

Climate change, including extreme variations in precipitation and prolonged extreme temperatures, could modify the population density of the dengue vector [2]. Consequently, variation in the dengue transmission risk may occur following Extreme Weather Events (EWEs), including heatwaves, drought, and flood [5–7]. Briefly, heatwaves, defined as prolonged periods of extreme heat over a given threshold, could result in higher mosquito populations as heatwaves can accelerate the early development stage of mosquitoes life cycle [8]. Drought conditions have been also linked to increased dengue transmission risk via water storage in man-made containers [9]. Moreover, while cumulative precipitation might enhance mosquito abundance, heavy precipitation flushes away breeding sites, thereby decreasing mosquito populations [10].

In the context of a warming climate, EWEs are anticipated to become more frequent and intense [11]. Predictive models suggest that the dengue-endemic SEA region is becoming more susceptible to the adverse effects of climate change that included but are not limited to heat stress, heavy precipitation, widespread flooding both inland and in coastal zones, prolonged dry spells and acute deficits in water supply [11]. Heatwave intensity is projected to rise by 0·5 to 1·5 °C above a given global warming threshold, with an increase in duration of 2–10 days annually per 1°C of global temperature rise [12]. Given that *Ae. aegypti* is the main vector of dengue fever in Southeast Asia, rising temperatures pose a real threat of increased dengue risk [13,14]. Climate models under Representative Concentration Pathway (RCP4·5) scenario and the 14-model ensemble mean from Coupled Model Intercomparison Project Phase 5 (CMIP5) projected an increasing trend of drought in SEA in the future [15].

Despite the substantial growth in evidence regarding the impacts of EWEs on human health, their effect on mosquito-borne diseases like dengue is limited, especially in SEA, where dengue is endemic [16]. In line with WHO's Global Vector Control Response operational framework, it is essential to investigate the impact of climate factors, particularly EWEs, on dengue using a multi-country approach in generating evidence required to efficiently prevent and manage dengue transmission and respond to subsequent outbreaks [17]. To our knowledge, this is the first study investigating the associations between multiple EWEs (i.e., heatwaves, extremely wet and drought conditions) and dengue in the SEA region, utilizing a robust statistical modelling approach.

## Methods

### Data collection

**Dengue data.** Monthly dengue count data were obtained from the Southeast Asia Research on Climate change and Dengue (SEARCD) collaboration platform for 268 locations in seven countries including Indonesia (33 provinces during 2010–2020), Lao PDR (2 provinces during 2005–2021), Malaysia (14 states and federal territories during 2010–2017), the Philippines (79 provinces during 2010–2020), Singapore (1 city state during 2010–2017), Thailand (76 provinces during 2003–2021), and Vietnam (63 provinces during 2011–2021). Briefly, SEARCD is a platform with collaborators from the seven SEA countries that encourages research on meteorological factors and dengue. Additionally, monthly dengue count data from 23 locations in Cambodia during 1998–2010 was obtained from project Tycho 2·0 [18]. The dengue dataset used in this study was fully anonymized prior to access and analysis. All identifying information was removed to ensure the privacy and confidentiality of individuals.

**Temperature and precipitation.** Monthly meteorological data, including mean temperature, mean dew point temperature, and total precipitation, were obtained from the ERA5-Land dataset generated by the European Centre for Medium-Range Weather Forecasts (ECMWF) (appendix pp. 2) [19]. ERA5-Land is a reanalysis data set providing climate information at a fine spatiotemporal resolution (0·1° × 0·1° grid) for the whole global land surface [19].

**Heatwave definitions.** Heatwaves were defined using relative thresholds, which consider the long-term daily mean temperature at each location. Heatwave were defined as two or more consecutive days with a daily mean temperature surpassing the 95th percentile of its distribution in each specific location [20,21]. Hourly temperature data were averaged to compute the daily mean temperature for each location. The total number of heatwave days (HWt) was then calculated for each month and location. If a heatwave begins in the final days of one month and continues into the next, the heatwave days are included to each respective month (S2 Table).

**Wet and drought conditions.** We used the Palmer Drought Severity Index (PDSI) to assess the effect of drought and extremely wet conditions on dengue. PDSI is a widely recognized standardized index for tracking drought and long-term variations in aridity and is computed by considering soil moisture content, the projected rate of evapotranspiration (i.e., the evaporation from soil under adequate water availability, considering mean daily temperature and numbers of days in the month) and rainfall amounts [22]. Here, we used the self-calibrating PDSI version (scPDSI) since it represents a geographically comparable index by calibrating a distinct normal condition for each location [23]. The scPDSI data were sourced at a geographical resolution of 0·5° × 0·5° grid from the Climatic Research Unit gridded Time Series (version 4·05) corresponding to the specific time in each location [24]. The scPDSI ranges from −10 (i.e., very dry conditions) to 10 (i.e., very wet conditions). Values below −4 or over 4 are categorized as drought and extremely wet conditions, respectively [23].

**Population data.** Total population count for each location were obtained from the Gridded Population of the World Version 4 developed by the Socioeconomic Data and Application Center for the period 2000–2020 at five-year intervals [25]. Linear interpolation was used to estimate yearly total population values for each location over the study period.

## Statistical analysis

We specified a generalized additive mixed model (GAMM) coupled with a distributed non-linear model (DLNM) to investigate the association between EWEs and dengue [26,27]. A quasi-Poisson distribution was assumed to account for potential over-dispersion in the dengue data. The logarithm of the total population for each location was included as an offset to adjust dengue case count by population. A smoothing spline with 5 degrees of freedom (df) per year was included to control for potential seasonality (S4 Table). A cross-basis function was included in each model using natural cubic spline with two equally spaced knots for heatwave and 3 df for scPDSI (i.e., the exposure dimension) and 3 df for the lag dimension based on exploratory analyses (S5 Table). We included a smoothing spline with 3 df to control for potential nonlinear effects of monthly mean temperature and total precipitation based on a previous study [28]. Long-term trends and interannual variability were accounted for incorporating an indicator term for each year in the time series. Unmeasured factors, such as public health interventions, were specified as unstructured random effects for each location. The reference value of heatwave was set at 0 heatwave days per month and scPDSI at 0. Models were fitted in R (version 4·3·1), using the packages *mgcv* 1·9·1 and *dlnm* 2·4·7 [26,27]. A flowchart summarizing the data collection and modeling steps was shown in S1 Fig.

The algebraic definition of the EWE-dengue model is given by:

$$\text{Log}(Y_{i,t}) = \alpha + \text{Log}(P_{i,t}) + f(\text{EWE}_{i,t}; l) + s(\text{Temp}_{i,t}) + s(\text{Preci}_{i,t}) + s(t) + \delta_{i,t} + \nu_i$$

where $Y_{i,t}$ indicates monthly dengue case at time t in location i, assumed to follow an over-dispersed quasi-Poisson distribution; $\alpha$ represents the intercept; $\text{Log}(P_{i,t})$ represents the logarithm of the total population per location i at the time t, included as an offset; $f(\text{EWE}_{i,t}; l)$ designate the cross-basis functions of either heatwave (HW) or scPDSI for maximum lag (l) of 4 months; $s(\text{Temp}_{i,t})$ and $s(\text{Preci}_{i,t})$ denotes the smooth functions of temperature and precipitation, respectively; $s(t)$ denotes the smooth seasonals trend of time; $\delta_{i,T}$ indicates long-term trend using indicator variables for each year T in each location i; and $\nu_i$ expresses unstructured random effects for each location.

## Sensitivity analysis

Multiple sensitivity analyses were carried out to assess the robustness of the results. We first examined the optimal exposure-response relationship in the cross-basis function by varying the df and placement of the knots, using natural cubic spline with two equally space internal knot, two knots at 25th and 75th percentile, and one knot at 50th percentile. We further applied a varying function of seasonality including smoothing spline with 3df, 4df, cyclic cubic spline, and one pair of Fourier term. Maximum lags were also varied by 3 months, 5 months, and 6 months, respectively. We repeated the analysis with adjustment relative humidity and population density in the model to account for additional confounding effects. Analyses using the data from 1998 to 2019 were also conducted to determine the potential impact of the COVID-19 pandemic on our results. Various heatwave definitions were investigated including: (i) four or more days exceeding the 95th percentile, (ii) two or more days exceeding the 97th percentile, (iii) two or more days exceeding the 99th percentile, (iv) four or more days exceeding the 97th percentile, and (v) four or more days exceeding the 99th percentile.

## Results

### Descriptive statistics

Table 1 shows the summary statistics of dengue cases and meteorological factors in each country. A total of 7,016,624 dengue cases was included in this study. The Philippines reported the highest mean annual dengue cases [196,719 cases (SD = 93,283)], followed by Vietnam [121,822 cases (SD = 81748)] and Indonesia [112,156 cases (SD = 42431)]. The monthly mean temperature, monthly total precipitation, mean annual heatwave days, and monthly scPDSI index at the country level ranged between 24·1°C and 26·8°C, 138mm and 254mm, 14·2 days and 17·2 days, and −0·3 and 0·3, respectively. The state of Selangor in Malaysia showed the highest monthly dengue incidence per 100,000 people followed by Khah Hoa in Vietnam (Fig 1A). The lower mean annual number of heatwave days was observed in south-eastern locations (Fig 1B). Eastern Philippines and Northern of Vietnam and Cambodia indicate wetter conditions (Fig 1C). Additional descriptive results of each climate variable and dengue incidence for each location are shown in S3 Table.

### Heatwave-dengue association

Fig 2A presents the pooled cumulative (lag 0–4 months) association between the total number of heatwave days per month and dengue compared to month without any heatwave day. The association between the monthly number of

**Table 1. Summary statistics of dengue cases, monthly temperature, total precipitation, heatwave days and scPDSI index in eight countries.**

|  | Period | No of locations | Total dengue case | Annual dengue case [mean (SD)] | Monthly mean temperature[a] (°C) [mean (SD)] | Monthly total precipitation[a] (mm) [mean (SD)] | Annual number of heatwave days[a,b] [mean (SD)] | Monthly scPDSI[a] [mean (SD)] |
|---|---|---|---|---|---|---|---|---|
| Cambodia | 1998–2010 | 23 | 152,058 | 11,697 (7,611) | 26·8 (0·9) | 151 (37) | 17·2 (2·4) | 0·3 (0·8) |
| Indonesia | 2010–2020 | 33 | 1,233,718 | 112,156 (42,431) | 24·7 (1·2) | 254 (51) | 14·2 (4) | 0·1 (0·6) |
| Lao PDR | 2005–2021 | 2 | 63,391 | 3,729 (4,793) | 25·1 (0·6) | 158 (14) | 17·1 (2·6) | 0·1 (0·6) |
| Malaysia | 2010–2017 | 14 | 545,479 | 68,185 (40,188) | 25·5 (0·8) | 222 (31) | 15·3 (5·9) | −0·3 (0·7) |
| Philippines | 2010–2020 | 79 | 2,163,905 | 196,719 (93,283) | 25·1 (1·4) | 230 (36) | 16·5 (4) | 0·8 (0·9) |
| Singapore | 2012–2019 | 1 | 91,463 | 11,433 (7,291) | 26·9 (0·6) | 207 (89) | 14·6 (2·9) | 0·8 (2·0) |
| Thailand | 2003–2021 | 76 | 1,426,568 | 75,083 (37,736) | 26·1 (1·2) | 138 (32) | 17 (2·3) | 0 (0·4) |
| Vietnam | 2011–2021 | 63 | 1,340,042 | 121,822 (81,748) | 24·1 (2·3) | 173 (26) | 15·2 (2·9) | 0 (0·8) |
|  | Total | 291 | 7,016,624 |  |  |  |  |  |

mm: millimeter; scPDSI stands for self-calibrated Palmer drought severity index; SD is standard deviation.

a : Location specific mean climate variable (SD).

b : 95th percentile for 2 consecutive days.

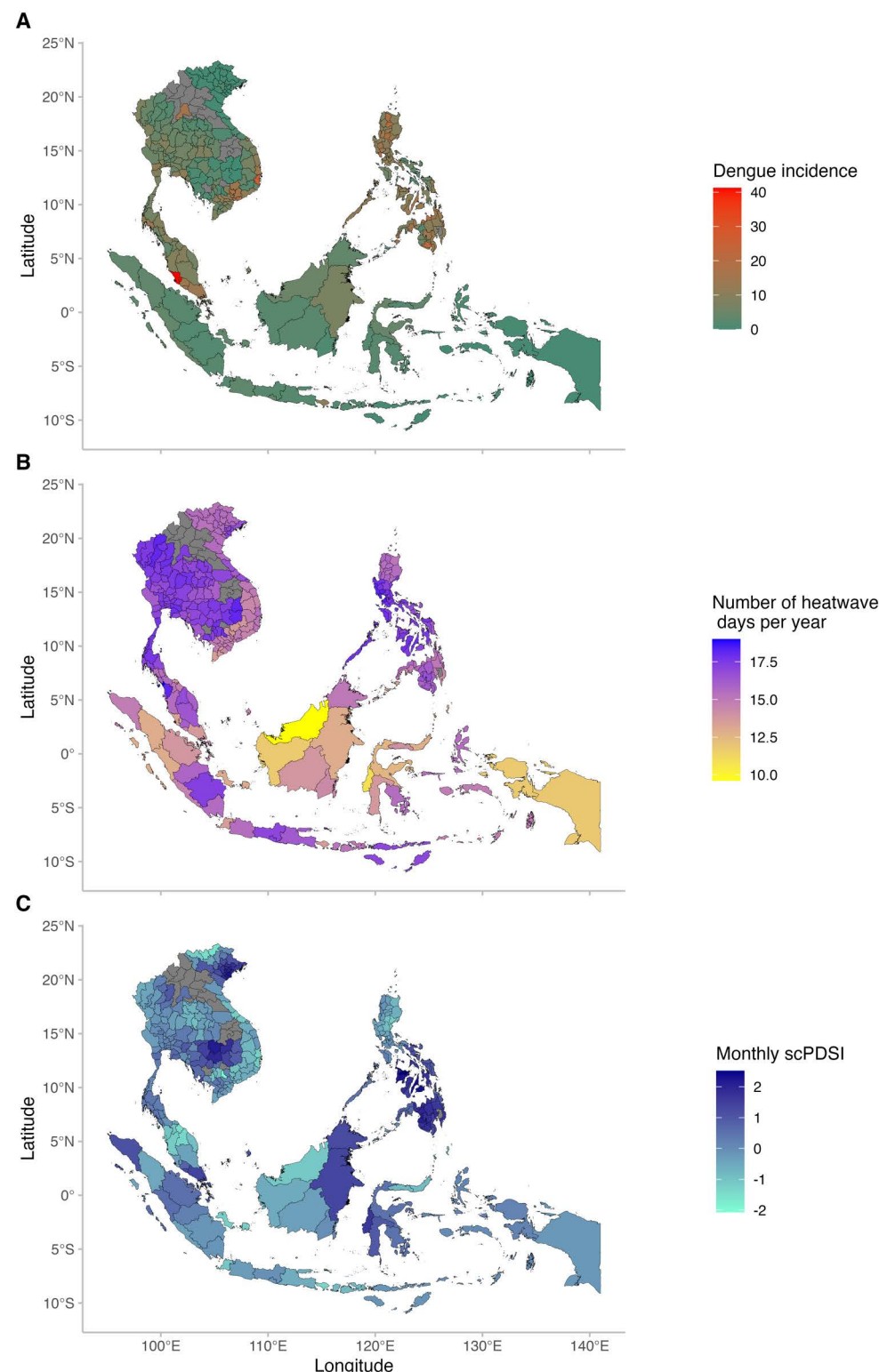

**Fig 1. Map of dengue case, heatwave, and scPDSI in each location of this study.** (A) Median monthly dengue incidence per 100,000 population in each location. (B) Mean annual number of heatwave days in each location. (C) Median monthly scPDSI index in each location. Grey areas have no data. scPDSI stands for self-calibrated Palmer drought severity index. Maps were created in R using Natural Earth data (https://www.naturalearthdata.com), accessed via the rnaturalearth R package.

heatwave days and dengue was non-linear and showed a significant positive association for months with less than 12 heatwave days and a negative association between months with 12 and 27 heatwave days. The dengue risk leveled off at months with 7 heatwave days (RR = 1·28; 95%CI: 1·19,1·38) and reached the minimum dengue risk at month with 21 heatwave days (RR = 0·49; 95%CI: 0·42, 0·57). Fig 2B indicates lag-response association curves of specific monthly number of heatwave days. Month with 7 and 21 heatwave days increase dengue risk after lag 2 and 3 months, respectively. The lag-response association showed a negative association across all numbers of heatwave days per month within a lag of two months (Fig 2C). However, the dengue risk increases mostly after the lag of two months.

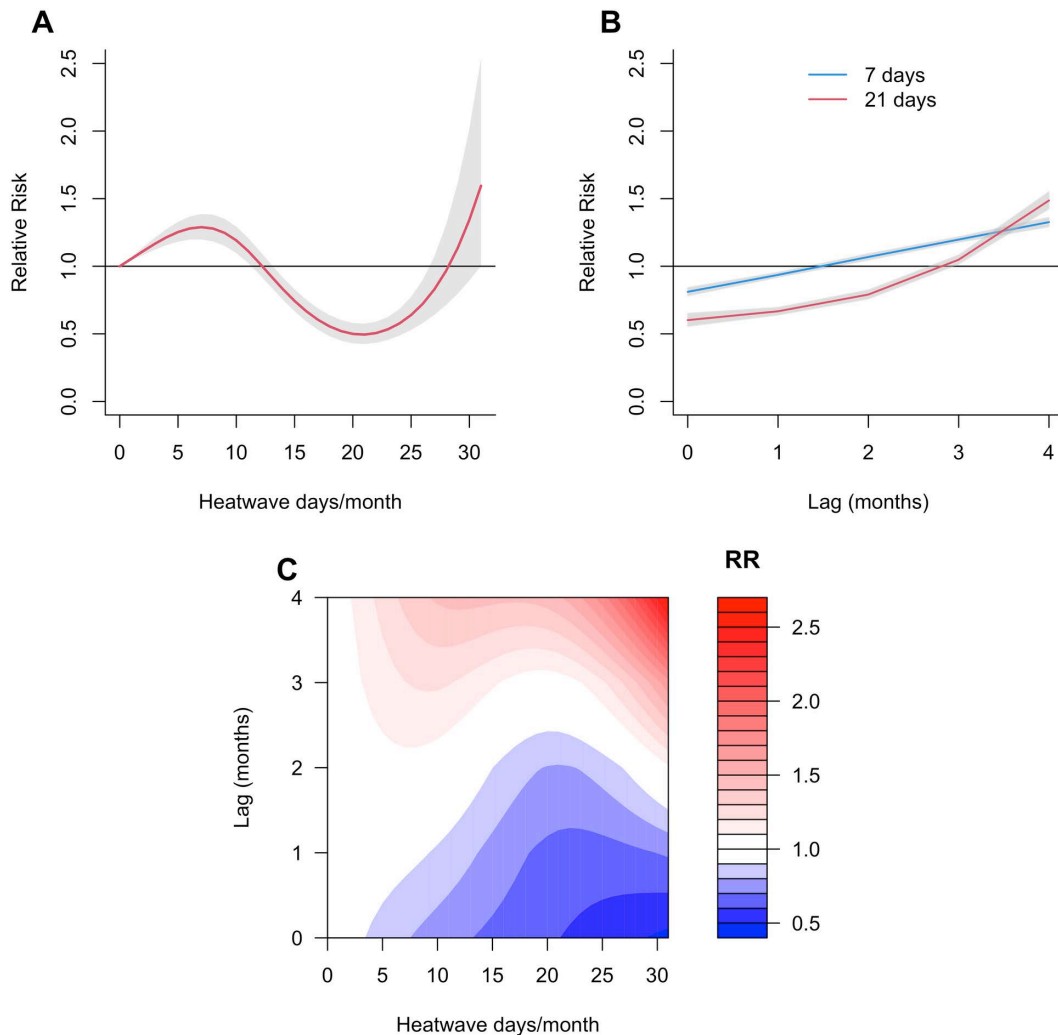

**Fig 2. Association between the number of heatwave days per month and dengue incidence in 291 locations across the eight countries, relative to month with no heatwaves.** (A) Cumulative exposure–response associations (with 95%CI, shaded grey), (B) lag-response association curves of specific monthly number of heatwave days (with 95% CI, shaded grey), (C) Contour plot of the association between monthly number of heatwave days and dengue across each lag. The deeper the shade of red, the greater the increase in the RR of dengue, and the deeper the shade of blue, the greater the decrease in the RR of dengue. CI indicates the confidence intervals; HW stands for number of heatwave days per month; RR is the relative risk.

## scPDSI-dengue association

Fig 3A shows the overall cumulative (lag 0–4 months) association between monthly scPDSI and dengue, compared to a value zero of scPDSI. The association between monthly scPDSI and dengue was non-linear and followed a reverse J-shaped curve association with minor upward tail and the minimum dengue risk observed at 2·5 scPDSI value (RR = 0·89; 95%CI: 0·87,0·91). Drought conditions (i.e. scPDSI values equal to or lower than −4) show a significant positive association with dengue incidence (RR = 1·85; 95%CI: 1·73,1·99). In contrast, the extremely wet conditions (i.e. scPDSI equal to 4) show a significant negative association with dengue risk (RR = 0·92; 95%CI: 0·87,0·96). An increase in dengue risk is observed when scPDSI is greater than 5·8, though the result is not significant.

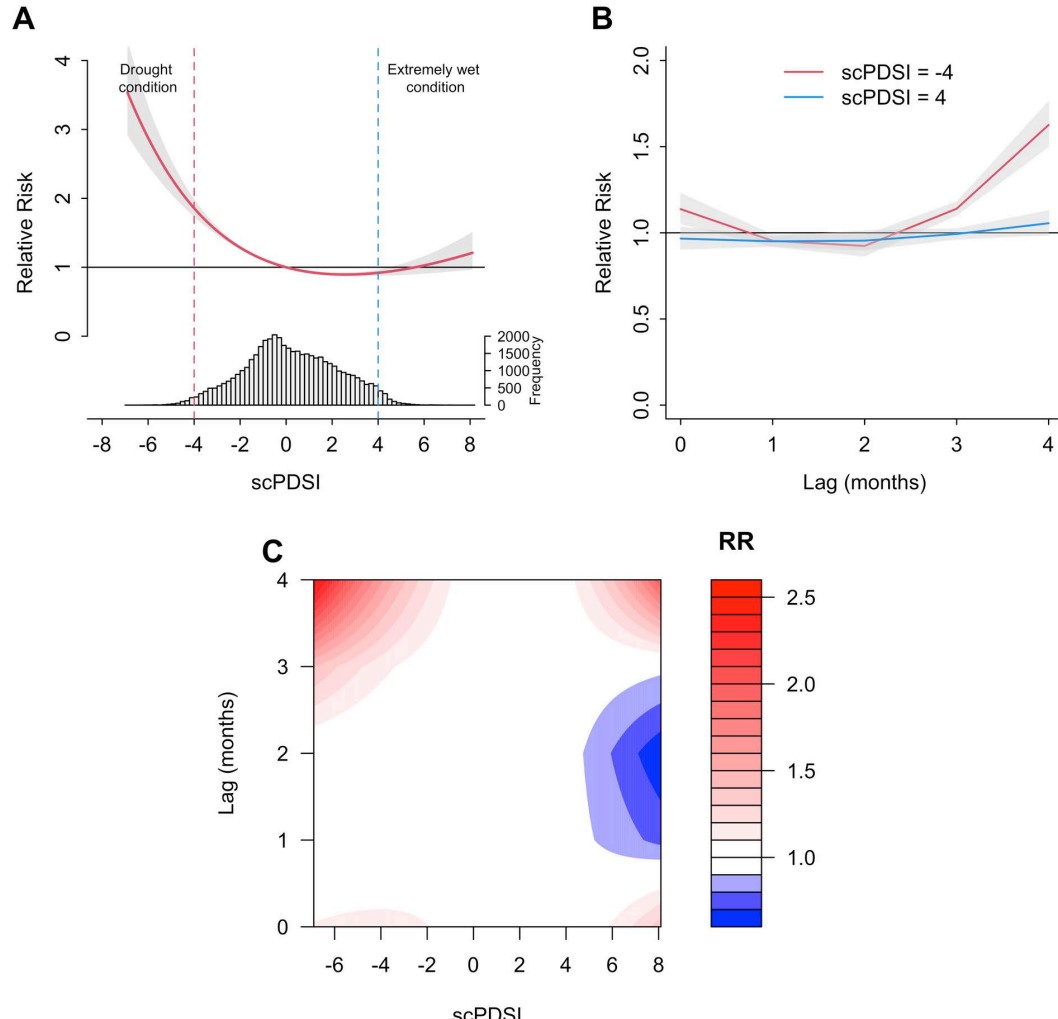

**Fig 3. Association between scPDSI and dengue incidence in 291 locations across the eight countries, relative to a scPDSI value of zero.** (A) Cumulative exposure–response associations (with 95%CI, shaded grey), with related PDSI distribution. Two vertical lines indicate the cut-off value for drought as a dotted line in red, and cut-off blue for extremely wet conditions as a dash-dotted line in blue. (B) lag-response association curves of specific scPDSI (with 95%CI, shaded grey). (C) Contour plot of the association between scPDSI and dengue across each lag. The deeper the shade of red, the greater the increase in the RR of dengue, and the deeper the shade of blue, the greater the decrease in the RR of dengue. CI indicates the confidence intervals; RR is the relative risk; scPDSI stands for self-calibrated Palmer drought severity index.

Fig 3B indicates association curves of specific scPDSI and dengue across all lag periods. Drought conditions (scPDSI ≤ −4) shows both immediate and delayed positive association on dengue risk within lag 0 and after two-months lag. Extremely wet conditions (scPDSI = 4) show an increased dengue risk at three-months lag, suggesting a delayed positive effect on dengue risk. Fig 3C illustrates the association between scPDSI and dengue risk across all lags. Notably, when the scPDSI exceeds 5·8, there is an increased dengue risk observed at a one month and after a three-month lag.

## Sensitivity analysis

The pooled effect estimates for each EWE on dengue incidence were generally robust, amidst varying adjustments for seasonality, maximum lag, as well as inclusion of relative humidity and population density (S8, S9, S10 Figs). Changing the df for the cross-basis function did not change the direction nor the intensity of the association between scPDSI and dengue but did change it for heatwaves (S7 Fig For the heatwave model, the results show a reduction in dengue relative risk regardless of the monthly number of heatwave days when two internal knots were used at the 25th and 75th percentile or one internal knot at the 50th percentile (S7 Fig). Analyses using the data from 1998 to 2019 shows no impact of the COVID-19 pandemic on our results (S11 Fig).

Using various heatwave definitions in the sensitivity analysis did not change the direction of the association, except for heatwave with the 99th percentile for 2 and 4 consecutive days (S12 Fig). With the definition of 99th percentile for 2 and 4 consecutive days, there was a negative association between zero to 16 heatwave days per month and dengue. This was likely because of significantly fewer counts of months with heatwave days when the strictest heatwave definition was used.

## Discussion

We examined the associations between several EWEs including heatwave, drought, and extremely wet conditions on dengue incidence in the SEA region. Months with fewer than 12 heatwave days showed increased dengue risks with a delayed effect observed after two months lag. This may partially be explained by the role of temperature on vector population since mosquitoes can survive even under continuous exposure to extreme heat amidst these heatwave durations [8]. In addition, heatwave events potentially facilitate vector population growth in the early development phase leading to an increased dengue transmission in the latter phase [8]. Also, the number of heatwave days may associate with a faster rate of viral replication within the vector and with a shorter extrinsic incubation period— the time required for dengue virus to become transmissible to another host after initial infection of a mosquito [2]. However, during the intense duration of heatwaves, adult mosquito survival rates and feeding activities start to decline, explaining our result of a reduction in dengue risk at month with more than 12 heatwave days [2]. In addition, while our study showed an increase in dengue risk following months with 27 heatwave days, the result is marginal due to the low number of months with an extreme frequency of heatwave days. This increase could potentially be attributed to the delayed effect of heatwaves on dengue risk. Further research is needed to investigate the impact of an extreme number of heatwave days in greater detail.

Our lag-exposure association result showed heatwaves more likely to increase dengue risk after a lag of two months. Comparing this finding with previous studies has its own limitations due to differences in heatwave definitions and statistical approaches used. However, a study in Hanoi, Vietnam, showed that heatwaves have an overall inhibitive effect at a shorter lag but were associated with increased magnitude of outbreaks at later lags [5]. In China, similar findings whereby heatwaves increase the risk of dengue outbreaks after six weeks were also observed [20]. This could be attributed to diapause behavior in mosquitoes, which is the inactive state where the dengue vectors halt their development and hatch to withstand harsh climatic conditions such as extreme heat or dry conditions [29]. The subsequent drops in temperature and increased humidity create favorable conditions for the continuation of the reproduction cycle of dengue vectors, ultimately leading to an increase in the dengue vector population [30]. The lags in mosquito outbreaks after a heatwave could result

from the influence of climatic fluctuations on mosquito biology, serving as an adaptation mechanism to cope with unfavorable climate conditions [30].

Additionally, the association may be related to the change in human activities during and after heatwaves. During the heatwave, people tend to stay indoors in air conditioned environment, which limits vector-host contact [5]. However, lower temperatures after heatwaves encourage people to spend more time outside, increasing the likelihood of being bitten by mosquitoes. Another explanation might be an impairment of the host immunological response after exposure to certain sustained high temperature during heatwave [31]. Prolonged heat stress can reduce the ability of human body to mount an effective immune response, potentially making it easier for the dengue virus to be viable as an infection [31].

scPDSI index and dengue has a non-linear association with drought conditions showing an increase in dengue risk and extremely wet conditions at scPDSI value between 4 and 5.8 associated with a reduction in dengue risk. A similar association was reported in other studies in Brazil and China [7,32]. We noted that drought was positively associated with dengue risk at lags 0 and after two months. This is likely because severe dry conditions increased water storage in artificial containers, contributing to an increase in reproduction sites and the proliferation of mosquitoes [9]. Mosquito eggs can remain viable for up to 120 days during drought conditions, hatching simultaneously when favorable conditions, such as the return of water availability, are restored [33]. The observed three-month lag between extreme drought events and an increase in dengue risk may stem from gradual changes in domestic water storage practices as households adapt to water scarcity. This change might prompt individuals to adopt measures such as storing water in makeshift containers around the home during periods of water shortage. This finding indicates the importance of ongoing monitoring of drought conditions and the prompt implementation of emergency measures to reduce preventable negative effects of dry conditions on dengue risk. For instance, practical strategies to address drought-related dengue risks include discouraging open water storage, sealing rooftop water systems, and covering household containers to minimize mosquito breeding and reduce the risk of dengue.

We found that extremely wet conditions showed a reduction in dengue risk, which is in line with a study in Singapore, where a significant reduction in dengue outbreak risk followed the flushing effect of precipitation [10]. Non-standing water could flush away the mosquito eggs and shelter, consequently decreasing the dengue risk [6]. However, our results showed that severely wet conditions increase dengue risk, which could potentially be due to stagnant waters amidst severe wet conditions favoring development and reproduction of mosquitoes [6]. Moreover, severely wet conditions may also contribute to hydrological natural disasters, including floods or tropical cyclones, leading to significant deterioration of infrastructure or electricity, increasing the risk of close contact between dengue vectors and humans [6]. However, the impact of severely wet conditions needs further investigation to understand this phenomenon.

## Strength and limitations

This research possesses a few strengths. First, this is the first study investigating of the EWEs-dengue association in the SEA region. This study was carried out in numerous subnational locations across most of the SEA region, representing different ranges of exposure levels and socio-demographic characteristics. By using a pooled design approach, this study provided strong evidence of the relationship between EWEs and dengue at the regional scale. Second, we used GAMM and DLNM that captures the complex non-linear and delayed relationships between climate variables and dengue incidence [26,27]. The use of GAMM enables the consideration of both spatial and temporal variations, which means the model can account for differences across locations and over time—factors that are critical when studying a vector-borne disease like dengue [26].

However, some limitations need to be acknowledged. First, a significant limitation is the use of a single set of modeling parameters across all the locations, which may heavily impact location/country-specific estimations. Specifically, the chosen modeling may not fit the data well in certain locations, even if it results in the best overall fit. However, the robustness of the results to various modeling choices and selections was assessed by exploring different model specifications for exposure and lag. Additionally, dengue case reporting can vary due to regional and temporal differences in diagnosis

approaches, leading to over- or under-reporting of dengue cases (S1 Table). These variations may complicate accurate assessments of dengue risk, potentially skewing understanding of the climate true impact on dengue. Moreover, utilizing monthly data to study dengue risks may not capture short-term climate effects and dengue incubation periods. Relying on monthly total heatwave days might overlook the nuances of individual heatwave episode impacts on dengue transmission. The potential influence of serotype information on the climate-dengue association is acknowledged. Due to data availability limitations, the analysis could not be stratified by dengue serotypes, highlighting the need for future research in this aspect. Due to the ecological design of our study, there is a potential for ecological fallacy, where the findings at the regional level may not accurately reflect the outcomes for individual countries or specific locations. Although factors such as urbanization and dengue control measures may influence or modify the impact of EWEs on dengue risk, we did not include these variables in our models as the study focused on the overall relationship between EWEs and dengue. Future research should explore how these factors modify the relationship between EWEs and dengue risk to provide a more comprehensive understanding of their interactions.

## Conclusions

Given the anticipated rise in the number and intensity of EWEs, this research offers insights into estimating the dengue risk associated with heatwave, drought, and extremely wet conditions introduced by the global climate in a real-world setting. Our study showed that delayed heatwaves and drought effects contribute to increased dengue risk. The findings highlight the need for public health interventions to mitigate the potential dengue risks posed by EWEs in the context of climate change in SEA.

## Supporting information

**S1 Appendix: Climate data collection.**
(DOCX)

**S1 Table. Summary information of dengue data for each country.**
(DOCX)

**S2 Table. Example calculation of monthly number of heatwave days (HWt).**
(DOCX)

**S3 Table. Location-specific summary table of mean and standard deviation of each variable of each country.**
(DOCX)

**S4 Table. Generalized cross-validation score of base dengue model with varying seasonality functions.**
(DOCX)

**S5 Table. Generalized cross-validation score of varying cross basis specification in main model.**
(DOCX)

**S6 Table. Pooled relative risks of heatwave-dengue association, relative to month with no heatwaves.**
(DOCX)

**S7 Table. The relative risks at each lag up to 4 months of heatwave-dengue association, relative to month with no heatwaves.**
(DOCX)

**S8 Table. Pooled relative risks of scPDSI-dengue association, relative to a scPDSI value of zero.**
(DOCX)

**S9 Table. The relative risks at each lag up to 4 months of scPDSI-dengue association, relative to a scPDSI value of zero.**
(DOCX)

**S1 Fig. Flowchart summarizing the data collection and modeling steps.**
(DOCX)

**S2 Fig. Map of the study area including 291 locations in eight Southeast Asia countries.**
(DOCX)

**S3 Fig. Time-series plot of total yearly dengue cases in eight Southeast Asia countries.**
(DOCX)

**S4 Fig. Pearson's correlation coefficients among dengue and climate variables.**
(DOCX)

**S5 Fig. Time series plot of monthly dengue and climate variables for the selective locations with the highest number of total dengue case for each country.**
(DOCX)

**S6 Fig. Heatwave-dengue association at different time lags (0–4 months).**
(DOCX)

**S7 Fig. scPDSI-dengue association at different time lags (0–4 months).**
(DOCX)

**S8 Fig. Sensitivity analysis result of varying cross-basis function for each heatwave-dengue, and scPDSI-dengue associations.**
(DOCX)

**S9 Fig. Sensitivity analysis result of using different seasonality functions for heatwave-dengue, and scPDSI-dengue associations.**
(DOCX)

**S10 Fig. Sensitivity analysis result of different maximum lag for heatwave-dengue, and scPDSI-dengue associations.**
(DOCX)

**S11 Fig. Sensitivity analysis result of adjusting for relative humidity and population density for heatwave-dengue, and scPDSI-dengue associations.**
(DOCX)

**S12 Fig. Sensitivity analysis result of conducting analysis with data from 1998–2019 (excluding covid year).**
(DOCX)

**S13 Fig. Sensitivity analysis result of different heatwave definitions.**
(DOCX)

**S14 Fig. Sensitivity analysis result of excluding Cambodia and Laos data.**
(DOCX)

## Acknowledgments

We would like to extend our sincere gratitude to Dr. Futoshi Hasebe and all collaborators for their valuable contributions to this study, who was supported by the Japan Agency for Medical Research and Development (AMED) (JP20wm0125006).

## Author contributions

**Conceptualization:** Sophearen Ith, Xerxes Seposo, Masahiro Hashizume.

**Data curation:** Sophearen Ith, Xerxes Seposo.

**Formal analysis:** Sophearen Ith.

**Funding acquisition:** Xerxes Seposo, Masahiro Hashizume.

**Investigation:** Sophearen Ith.

**Methodology:** Sophearen Ith, Xerxes Seposo, Alex R Cook, Felipe J Colón-González, Masahiro Hashizume.

**Resources:** Xerxes Seposo, Kraichat Tantrakarnapa, Geminn Louis C. Apostol, Pandji Wibawa Dhewantara, Rozita Hod, Mohd Rohaizat Hassan, Hidayatulfathi Othman, Mazrura Sahani, Jue Tao Lim, Ha Hong Nhung, Nguyen Hai Tuan, Ngu Duy Nghia, Taichiro Takemura, Inthavong Nouhak, Paul L. C. Chua, Masahiro Hashizume.

**Software:** Sophearen Ith, Vitou Phy.

**Supervision:** Xerxes Seposo, Masahiro Hashizume.

**Validation:** Sophearen Ith.

**Visualization:** Sophearen Ith.

**Writing – original draft:** Sophearen Ith.

**Writing – review & editing:** Sophearen Ith, Xerxes Seposo, Vitou Phy, Kraichat Tantrakarnapa, Geminn Louis C. Apostol, Pandji Wibawa Dhewantara, Rozita Hod, Mohd Rohaizat Hassan, Hidayatulfathi Othman, Mazrura Sahani, Jue Tao Lim, Ha Hong Nhung, Nguyen Hai Tuan, Ngu Duy Nghia, Taichiro Takemura, Inthavong Nouhak, Paul L. C. Chua, Alex R Cook, Felipe J Colón-González, Masahiro Hashizume.

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
