## [Decision Letter · Decision Letter 0]

7 Jan 2025

PNTD-D-24-01566

Extreme weather events and dengue in Southeast Asia: a regionally-representative analysis of 291 locations from 1998 to 2021

Dear Dr. Seposo,

Thank you for submitting your manuscript to PLOS Neglected Tropical Diseases. After careful consideration, we feel that it has merit but does not fully meet PLOS Neglected Tropical Diseases's publication criteria as it currently stands. Therefore, we invite you to submit a revised version of the manuscript that addresses the points raised during the review process.

Please submit your revised manuscript within 60 days Mar 08 2025 11:59PM. If you will need more time than this to complete your revisions, please reply to this message or contact the journal office at plosntds@plos.org. Please include the following items when submitting your revised manuscript:

We look forward to receiving your revised manuscript.

Kind regards,

Mohamed Gomaa Kamel

Academic Editor

Nigel Beebe

Section Editor

Shaden Kamhawi

co-Editor-in-Chief

Paul Brindley

co-Editor-in-Chief

**Journal Requirements:**

1) Please provide an Author Summary. This should appear in your manuscript between the Abstract (if applicable) and the Introduction, and should be 150-200 words long. The aim should be to make your findings accessible to a wide audience that includes both scientists and non-scientists. Sample summaries can be found on our website under Submission Guidelines:

Potential Copyright Issues:

- Figures 1 and S1; Please provide a direct link to the base layer of the map (i.e., the country or region border shape) and ensure this is also included in the figure legend; and provide a link to the terms of use / license information for the base layer image or shapefile. We cannot publish proprietary or copyrighted maps (e.g. Google Maps, Mapquest) and the terms of use for your map base layer must be compatible with our CC BY 4.0 license.

**Reviewers' Comments:**

Reviewer's Responses to Questions

**Key Review Criteria Required for Acceptance?**

**Methods**

-Are the objectives of the study clearly articulated with a clear testable hypothesis stated?

-Is the study design appropriate to address the stated objectives?

-Is the population clearly described and appropriate for the hypothesis being tested?

-Is the sample size sufficient to ensure adequate power to address the hypothesis being tested?

-Were correct statistical analysis used to support conclusions?

-Are there concerns about ethical or regulatory requirements being met?

Reviewer #1: The objectives and tested hypothesis are clearly articulated.The study design and sample size (>7 MM Dengue cases) apppropriate and robust. The population description and appropriateness has the usual limitations eg reporting bias of only severe vs severe +non severe dengue cases acknowledged by the authors.

There are no ethical nor regulatory issues.

The statistical modeling methods used are the essential tools and needs to be validated by reviewer(s) / experts in these matters ie statisticians /modelers..as I am not enough versed in these two critical areas.

Reviewer #2: The objectives of the study clearly articulated.

The study design appropriate to address the stated objectives.

The population is clearly described and appropriate for the hypothesis being tested.

Correct statistical analysis used to support conclusions.

There are no concerns about ethical or regulatory requirements.

Reviewer #3: The aim of the study was to verify the association between extreme weather events and dengue transmission in Southeast Asia.

The association between dengue incidence with three extreme weather variables (heatwaves, droughts and extereme wet conditions) was analyzed in an ecologic design, using different time lags.

The "population" of the study was composed of 291 administrative and geographic divisions of seven countries located in the region.

The study used publicly available data from disease surveillance systems and weather variables.

A GAMM model was used to fit the data.

Reviewer #4: (No Response)

Reviewer #5: This reviewer gives a mixed assessment on the methods. Several strengths deserve commendation: (1) The authors have assembled a rich data set on dengue cases spanning multiple countries, many decades, and monthly level data. (2) They have applied a sophisticated statistical methodology; (3) The approach appears to be well executed. In addition, their thorough review of the literature deserves praise.

On the other hand, the methods seem to have some limitations. The greatest concern is the absence of considering the "multiple" comparisons problem. Classical significance tests fit well to a single, pre-specified test. The methods in this paper did not indicate which, if any, parts of the specification had been chosen before the authors started their analysis, and how their revised their approach based on their interim results. As is well known, the more tests that are performed, the greater the chance that some "significant" result can be found by chance. In this study, apparent opportunities for multiple comparison could arise in many ways, such as: (1) The choice of the statistical model. The authors chose smoothed splines. They could have chosen averages, a specific form, a model incorporating the month of the year as an independent variable, or some other structure. (2) The specific parameters within the chosen model. For example, the spline used 5 degrees of freedom. The sensitivity analyses considered a small limited range but not a full defense against multiple comparisons; (3) The definition of the heat index. The authors give citations 18 and 19, but do not say whether they explicitly followed those prior definitions. (4) The drought index. While the use of an established index is a virtue, they authors did not give a prior endorsement as to why an index developed for agriculture would apply to dengue risk.

Additional concerns relate to the exclusion of potentially relevant variables. The authors explicitly acknowledged the exclusion of dengue control measures, which might have been very relevant However, they also excluded time and items associate with it, such as increasing national per capita income, degree of urbanization (as dengue is a primarily urban disease, and variations among sites. The authors' hypothesis around drought responses would seem especially affected by these omitted variables. Higher income and more urban environments might be more likely to use air conditioning or have household members spending time in such settings working, studying, or shopping. Likewise, storing water in unsealed containers would apply most to locations mostly in lower income countries.

If the authors were able to incorporate some of these variables in their models their paper would be strengthened. If they cannot, they may wish to acknowledge that limitation and mention some of the topics as promising areas for further research.

Reviewer #6: -Are the objectives of the study clearly articulated with a clear testable hypothesis stated?

Answer Yes

-Is the study design appropriate to address the stated objectives?

Answer Not sure

-Is the population clearly described and appropriate for the hypothesis being tested?

Answer Yes

-Is the sample size sufficient to ensure adequate power to address the hypothesis being tested?

Answer Yes

-Were correct statistical analysis used to support conclusions?

Answer Not sure

-Are there concerns about ethical or regulatory requirements being met?

Answer Not sure

**Results**

-Does the analysis presented match the analysis plan?

-Are the results clearly and completely presented?

-Are the figures (Tables, Images) of sufficient quality for clarity?

Reviewer #1: Yes

Yes

The figures lack simplicity for a non expert reader. Maybe some clarifications could be made

Reviewer #2: The analysis presented match the analysis plan.

The results are clearly and completely presented.

Figures are of sufficient quality for clarity?

Reviewer #3: Yes, the analysis matches the analysis plan. In my view the tables and images are clear.

Reviewer #4: (No Response)

Reviewer #5: Generally satisfactory. However, it was not clear why the variability in dengue reporting was a factor. Passive dengue reporting is known to capture only a portion of dengue cases. Generally the proportion increases with the quality of the country's health system (see papers by Shepard DS, Undurraga E and colleagues). But if that fraction were consistent over time, it would be captured in the location-specific constant.

Reviewer #6: -Does the analysis presented match the analysis plan?

Answer The analysis presented partially aligns with the analysis plan.

-Are the results clearly and completely presented?

Answer No

The analysis presented partially aligns with the analysis plan.

Answer No

**Conclusions**

-Are the conclusions supported by the data presented?

-Are the limitations of analysis clearly described?

-Do the authors discuss how these data can be helpful to advance our understanding of the topic under study?

-Is public health relevance addressed?

Reviewer #1: yes to all.

I think the summary conclusion "Our finding offer stakeholders sizeable amount of time to organize and implement public health interventions in minimizing the prospective dengue risks" need to be rewritten in a more rigorous way underscoring the quantitative study outcomes.

Reviewer #2: The conclusions are supported by the data presented.

The limitations of analysis are clearly described.

The authors discuss how these data can be helpful to advance our understanding of the topic under study.

Public health relevance is addressed.

Reviewer #3: The limitations related to the ecologic design should be acknowledged.

Reviewer #4: (No Response)

Reviewer #5: Despite the noted strengths, this reviewer is concerned about the effect of the limitations in the methods on the conclusions. Trying to slow long term global warming may be a long term approach. Based on the study's complex relationships, it would be helpful to summarize the next impact of global warming on dengue cases.

I would encourage the authors to clarify their purpose of their study by inserting the word "cases" after "dengue" and the adjective "exploratory) before the words "regionally- representative..." in the title of their manuscript.

Reviewer #6: -Are the conclusions supported by the data presented?

Answer Partly

-Are the limitations of analysis clearly described?

Answer Partly

-Do the authors discuss how these data can be helpful to advance our understanding of the topic under study?

Answer Partly

-Is public health relevance addressed?

Answer No

**Editorial and Data Presentation Modifications?**

Reviewer #1: Confirm the contributions of the impressive co-authors list ie from 18 institutions vs acknowledgments..

Reviewer #2: Minor revision

Reviewer #3: (No Response)

Reviewer #4: (No Response)

Reviewer #5: line 59. Change "Flaviviruses" to lower case.

line 80. After "2 to 10 days" insert the time period (e.g. annually).

line 84. After "limited," insert a comma.

line 139. Inset "a" before "previous study"

line 172. Insert "The" before "State of Selangor

line 178. Column for precipitation. Add units to label (e.g. mm)

line 212 Add "the" before result and change "result" to "results."

line 226 Change "event" to "events"

line 278 Change "We" to lower case "we"

line 319 Change" Researcher" to "Researchers"

Reviewer #6: Minor Revision

**Summary and General Comments**

Reviewer #1: Study is novel, supported by a large data set, statistical/modeling methods need to be validated by experts in these areas, Confirm the contributions of the impressive co-authors list ie from 18 institutions vs acknowledgments for publication ethics.

Discussion is well balanced and the conclusions are impactful.

Reviewer #2: This study was very well presented. The study’s idea was also clear and was practical for epidemiological and community health purpose. I can see that the authors did many attempts in statistical analyses to check every possibilities those might affect the results of the study. The results were also explained well in the discussion section.

To improve this article, there are some issues authors need to elaborate. Those are:

1. Table 1: data for each country covered different time period, e.g. Cambodia 1998-2010 while Indonesia 2010-2020. As we all know, ideally data must be collected in similar time period. Different timing in data collection may potentially lead to ’temporal bias’. Is there any analysis/ strategy authors provided to overcome this period differences?

2. Different mosquito species have different characteristic. From a literature (http://www.parasitesandvectors.com/content/7/1/338), we could learn that temperature does not limit the persistence of Ae. albopictus. In this study, authors analysed the dengue incidence in association with selected EWEs. Providing data about the dominant mosquito species in each region will be a valuable addition to the explanation/ discussion.

3. Is it possible that each country gave different result when analysed separately? As each country might have different weather variability and different vector control strategy (e.g. some countries apply ‘closing, draining, and burying water-filled container’. This behaviour might reduce dengue incidence even in drought/ heatwave).

Reviewer #3: Extreme weather events have become more frequent. They have occurred on a worldwide scale. Analysis of its association with the occurrence of vector-borne diseases should be encouraged. However, local factors may have significant importance in this relationship.

Reviewer #4: This manuscript investigates the associations between extreme weather events (EWEs)—heatwaves, droughts, and extremely wet conditions—and dengue incidence across Southeast Asia. Using extensive data spanning over two decades from multiple locations, the study employs advanced statistical methodologies, including a generalized additive mixed model (GAMM) combined with a distributed lag non-linear model (DLNM). It addresses a timely and critical public health issue in the context of climate change, demonstrating methodological rigor, integrating diverse datasets, and offering actionable insights for policymakers. Additionally, the study carefully considers and excludes potential confounding influences, such as COVID-19. However, the manuscript would benefit from addressing the following issues:

1. The dengue data collection periods vary across countries, with some data not covering the full span from 1998 to 2021. For instance, only two provinces in Lao PDR are included. This raises questions about potential biases or limitations in the results. The manuscript should clarify the rationale behind these inconsistencies and assess their impact on the study's findings.

2. While the methodological details are well-presented, the inclusion of a flowchart summarizing the data collection and modeling steps would enhance transparency and aid reader comprehension.

3. Prolonged heatwaves often lead to drought conditions. The manuscript should explore whether there is any overlap or interaction between these two EWEs and how this might influence dengue risk.

4. The descriptions of **Figure 2B**, **Figure 3B**, and **Figure 3C** (Page 9, lines 188-190, 202-206; Page 11, lines 245-255) appear unclear or inaccurate. The manuscript would benefit from specifying and clearly labeling the coordinate axes to improve interpretability.

5. On Page 10-11, lines 230-232, the manuscript explains the decline in dengue risk during months with more than 12 heatwave days due to reduced mosquito survival and feeding activity. However, the subsequent increase in dengue risk during months with over 27 heatwave days is not adequately addressed. The authors should provide a plausible explanation for this observation.

6. The statement on Page 11, line 253, that "extremely wet conditions show a reduction in dengue risk" is misleading. An increase in dengue risk is observed when the scPDSI exceeds 5.8, even if the result is not statistically significant. This should be rephrased for accuracy.

7. On Page 12, lines 257-258, the manuscript notes that mosquito eggs can survive up to 120 days in drought conditions. However, Figure 3B shows a rising relative risk (RR) of dengue after a lag of more than three months, with greater increases over time. The authors should reconcile these findings and provide a detailed explanation.

This manuscript has the potential to significantly advance our understanding of the complex interactions between climate change and dengue risk. Addressing the outlined issues will enhance its clarity, robustness, and practical implications.

Reviewer #5: Besides areas for further research, the authors' conclusions should indicate implications for mitigating climate risks, such as avoiding the storage of water or ensuring that rooftop systems are airtight and household containers are covered to avoid breeding mosquitoes.

Reviewer #6: This study is interesting, featuring a large and diverse population spanning 291 locations across eight Southeast Asian countries. It incorporates sensitivity analysis, considers confounding factors, and takes into account the impact of COVID-19. However, there are some points of concern and recommendations for improvement as follows:

1. Rationale for Selecting the Models: This study utilized functions to control environmental factors, considering various related factors such as seasonality, environmental influences, and random variables. However, the rationale for selecting this model raises the question of how confident we can be that the chosen model (GAMM) is the optimal one. To determine the optimal analytical methodology for examining the association between EWEs and dengue for this dataset, how can this confidence be justified?

2. Smoothing Spline with 3 df: Your study employs a smoothing spline with 3 degrees of freedom based on a previous study conducted in Yogyakarta Province, Indonesia, which used Generalized Linear Regression Models (DOI:10.1371/journal.pone.0152688). Additionally, alternative methods, such as cross-basis functions, were also applied. How confident are you that these methods are the most appropriate for the dataset used in this study?

3. Definition of Heatwave: How confident can you be that the definition of heatwave used in this study aligns with the conditions in the areas studied and effectively represents its actual impact across all locations?

4. Managing Overfitting: How did you address the impact of overfitting when conducting multiple sensitivity analyses?

5. Statistical Reporting: The statistical reporting should include measures of size and/or distribution. For example, in lines 168-169, only the mean annual dengue case count is shown. These details should be more comprehensive.

6. Assessment Results: The results from the assessment between the independent and dependent variables should comprehensively present the magnitude and distribution while clearly indicating statistical significance levels.

7. Reporting Dengue Cases: Why was only RR used to report dengue case numbers?

8. Consistency of Terminology: Ensure consistent use of terms. For example, in line 184, the term "dengue relative" is used, in line 194, "only dengue," and in line 203, "dengue risk." Consistency in terminology would enhance clarity.

9. Abstract and Conclusions: The abstract and conclusions should be clearer and more robust. Recommendations for future research should build on this study, and practical implications for dengue prevention and control should be more explicitly addressed.

10. Appendix Formatting: Table headers should be included on every page of the appendix.

11. I do not see any mention of the ethical considerations in the manuscript. I am not certain whether any identifiable personal data was collected (even if it was not utilized or reported in this study). Since this study involves data collected from multiple countries, if no identifiable personal data or other ethically sensitive information was collected, it should be explicitly stated in the manuscript.

PLOS authors have the option to publish the peer review history of their article (what does this mean? ). If published, this will include your full peer review and any attached files.

**Do you want your identity to be public for this peer review?** For information about this choice, including consent withdrawal, please see our Privacy Policy .

Reviewer #1: **Yes: ** Jean Lang MD PhD FASTMH

Reviewer #2: **Yes: ** Leonard Nainggolan

Reviewer #3: **Yes: ** Expedito Luna

Reviewer #4: No

Reviewer #5: **Yes: ** Donald S. Shepard, PhD

Reviewer #6: No

**Figure resubmission:**

**Reproducibility:**



---

## [Decision Letter · Decision Letter 1]

8 Aug 2025

Dear Dr. Seposo,

We are pleased to inform you that your manuscript 'Extreme weather events and dengue in Southeast Asia: a regionally-representative analysis of 291 locations from 1998 to 2021' has been provisionally accepted for publication in PLOS Neglected Tropical Diseases.

Best regards,

Nigel Beebe, PhD

Section Editor

Shaden Kamhawi

co-Editor-in-Chief

Paul Brindley

co-Editor-in-Chief

Reviewer's Responses to Questions

**Key Review Criteria Required for Acceptance?**

**Methods**

-Are the objectives of the study clearly articulated with a clear testable hypothesis stated?

-Is the study design appropriate to address the stated objectives?

-Is the population clearly described and appropriate for the hypothesis being tested?

-Is the sample size sufficient to ensure adequate power to address the hypothesis being tested?

-Were correct statistical analysis used to support conclusions?

-Are there concerns about ethical or regulatory requirements being met?

Reviewer #2: I do not have any further comments on the methods. Methods has been covered very well by Reviewer #5.

Reviewer #3: The authors have properly addressed the issues raised by the reviewers.

Reviewer #4: (No Response)

Reviewer #6: -Are the objectives of the study clearly articulated with a clear testable hypothesis stated?

ANS Yes

-Is the study design appropriate to address the stated objectives?

ANS Partially, While the study’s regional scope and focus on climate-related determinants of dengue incidence are well-aligned with the stated objectives, there are critical gaps in the research design that limit its ability to fully address those objectives.

-Is the population clearly described and appropriate for the hypothesis being tested?

ANS Yes

-Is the sample size sufficient to ensure adequate power to address the hypothesis being tested?

ANS YES

-Were correct statistical analysis used to support conclusions?

ANS Partially.The selection of the Generalized Additive Mixed Model (GAMM) appears appropriate given the study’s intent to capture potentially non-linear associations between environmental variables (e.g., heatwave, drought) and dengue incidence. However, several important statistical considerations remain unaddressed.

-Are there concerns about ethical or regulatory requirements being met?

ANS No

**Results**

-Does the analysis presented match the analysis plan?

-Are the results clearly and completely presented?

-Are the figures (Tables, Images) of sufficient quality for clarity?

Reviewer #2: Results have been clearly presented. It has addressed my previous comments about the different time period of data collection.

Reviewer #3: The authors have properly addressed the issues raised by the reviewers.

Reviewer #4: (No Response)

Reviewer #6: -Does the analysis presented match the analysis plan?

ANS Yes

-Are the results clearly and completely presented?

ANS Partially

-Are the figures (Tables, Images) of sufficient quality for clarity?

ANS Partially

**Conclusions**

-Are the conclusions supported by the data presented?

-Are the limitations of analysis clearly described?

-Do the authors discuss how these data can be helpful to advance our understanding of the topic under study?

-Is public health relevance addressed?

Reviewer #2: Conclusion supported the data presented.

Reviewer #3: The authors have properly addressed the issues raised by the reviewers.

Reviewer #4: (No Response)

Reviewer #6: -Are the conclusions supported by the data presented?

ANS Yes

-Are the limitations of analysis clearly described?

ANS Partially

-Do the authors discuss how these data can be helpful to advance our understanding of the topic under study?

ANS Yes

-Is public health relevance addressed?

ANS Yes

**Editorial and Data Presentation Modifications?**

Reviewer #2: Authors has addressed my other comments regarding the role of different mosquito species and the possibility of analysing each individual location.

In the 1st sentence of introduction (line 75 of manuscript): “Dengue is caused for flaviviruses, transmitted by the infectious bite of Aedes aegyti, and Ae. albopictus mosquitoes.”

I suggest that it is better if the phrase “caused for” change to “caused by” or “caused from”.

Reviewer #3: The authors have properly addressed the issues raised by the reviewers.

Reviewer #4: (No Response)

Reviewer #6: Minor Revision

**Summary and General Comments**

Reviewer #2: The authors have addressed all the comments from the reviewer (either into revision to the article or into explanation to the reviewer).

Reviewer #3: The authors have properly addressed the issues raised by the reviewers.

Reviewer #4: (No Response)

Reviewer #6: This study presents valuable contributions to the field, particularly in its aim to examine dengue epidemiology across the Southeast Asian (SEA) region. By covering data from seven countries and 268 locations (including 23 in Cambodia), the study provides a potentially impactful dataset that could inform the design of region-specific public health interventions for dengue prevention and control, as well as serve as a reference for future research.

However, several critical issues must be addressed to enhance the validity and reliability of the findings. These concerns are mainly related to two major components of the research process: the data collection process and the statistical methodology.

1. Data Collection Process

The study identifies heatwaves and humidity/drought conditions as key independent variables, while dengue incidence is the primary outcome. While several aspects are discussed in the manuscript, the following key points require further clarification or improvement:

1.1 Variable Definitions and Regional Heterogeneity

Given the regional scope, the study must ensure consistency and clarity in the definition and operationalization of key variables, particularly across diverse national contexts. For example:

- The operational definitions of “heatwave” and “drought” should be contextually adapted to local climatic and public health standards.

- The study should clarify whether these thresholds have been validated or shown to have empirical effects on dengue transmission.

In addition, data sources should be validated repositories, and the authors must account for potential biases, such as inconsistent reporting mechanisms (e.g., passive surveillance systems) and national differences in data availability. Furthermore, potential confounding factors—notably urban versus rural setting, population density, and socioeconomic factors—must be considered, as they are known to influence dengue dynamics.

1.2 Data Quality Control and Ethics Internal validation procedures such as double data entry or cross-referencing across multiple datasets would strengthen data reliability. Furthermore, the manuscript lacks any mention of Institutional Review Board (IRB) approval or ethical clearance procedures. IRB review is essential not only for data anonymization but also for ensuring transparency, data acquisition appropriateness, and ethical integrity, especially when dealing with health surveillance data.

2. Statistical Methods Assuming that the data collection process is refined as above, attention must also be given to the statistical modeling and interpretation:

2.1 Model Selection and Comparison The use of a Generalized Additive Mixed Model (GAMM) is appropriate for capturing non-linear relationships. However, model selection should be justified through comparison with alternative models (e.g., GLMM, GEE) using statistical criteria such as AIC, BIC, or cross-validation errors. This would demonstrate that the chosen model is not just reasonable but optimal.

2.2 Control of Type I Error in Multiple Testing Given the number of variables and comparisons performed, it is important to control for multiple comparisons using appropriate techniques, such as Bonferroni correction or Benjamini-Hochberg FDR control, to avoid inflated false-positive rates.

2.3 Confounding Variables Variables such as urbanization, access to healthcare, and socioeconomic status are known to mediate both exposure (heat/humidity) and outcome (dengue). Failing to account for these may lead to spurious associations. Their inclusion as covariates or the use of hierarchical or multilevel modeling would improve the robustness and generalizability of the findings.

2.4 Diagnostics and Model Assumptions The manuscript lacks discussion of model diagnostics (e.g., residual checks, overdispersion, autocorrelation), which are crucial for validating the assumptions of the GAMM framework.

3. Implications of Data Validity

If the data collection process is flawed—e.g., lacking standardization or containing systematic errors—then the indicators such as heatwaves or drought may be inaccurately correlated with dengue incidence. This misalignment may lead to erroneous conclusions, compromising the effectiveness of future public health interventions. Misguided policy decisions based on unreliable evidence can result in the misallocation of resources, public health costs, and reduced trust in surveillance-driven research. Moreover, it undermines the foundation for future investigations, potentially skewing the trajectory of research and policy development in vector-borne disease control.

PLOS authors have the option to publish the peer review history of their article (what does this mean? ). If published, this will include your full peer review and any attached files.

**Do you want your identity to be public for this peer review?** For information about this choice, including consent withdrawal, please see our Privacy Policy .

Reviewer #2: **Yes: ** Leonard Nainggolan, MD, PhD

Reviewer #3: **Yes: ** Expedito Luna

Reviewer #4: **Yes: ** Ran Wang

Reviewer #6: No

---

## [Editor Report · Acceptance letter]

Dear Dr. Seposo,

We are delighted to inform you that your manuscript, " 

Extreme weather events and dengue in Southeast Asia: a regionally-representative analysis of 291 locations from 1998 to 2021," has been formally accepted for publication in PLOS Neglected Tropical Diseases.

Best regards,

Shaden Kamhawi

co-Editor-in-Chief

Paul Brindley

co-Editor-in-Chief
